# Novel estimates reveal subnational heterogeneities in disease-relevant contact patterns in the United States

**Casey F. Breen** *, **Ayesha S. Mahmud**, **Dennis M. Feehan**

Department of Demography, University of California, Berkeley, Berkeley, California, United States of America

* caseybreen@berkeley.edu

## Abstract

Population contact patterns fundamentally determine the spread of directly transmitted airborne pathogens such as SARS-CoV-2 and influenza. Reliable quantitative estimates of contact patterns are therefore critical to modeling and reducing the spread of directly transmitted infectious diseases and to assessing the effectiveness of interventions intended to limit risky contacts. While many countries have used surveys and contact diaries to collect national-level contact data, local-level estimates of age-specific contact patterns remain rare. Yet, these local-level data are critical since disease dynamics and public health policy typically vary by geography. To overcome this challenge, we introduce a flexible model that can estimate age-specific contact patterns at the subnational level by combining national-level interpersonal contact data with other locality-specific data sources using multilevel regression with poststratification (MRP). We estimate daily contact matrices for all 50 US states and Washington DC from April 2020 to May 2021 using national contact data from the US. Our results reveal important state-level heterogeneities in levels and trends of contacts across the US over the course of the COVID-19 pandemic, with implications for the spread of respiratory diseases.

**Data Availability Statement:** All code and data used to produce the analyses presented in this manuscript are available from an Open Science

## Author summary

Reliable quantitative estimates of contact patterns are in high demand because they are a critical input for modeling and reducing the spread of directly transmitted infectious diseases such as COVID-19 and influenza. But, while national-level contact data are available from surveys collected in many countries—including the United States—local-level estimates of age-specific contact patterns are almost never available. Yet these local-level contact patterns are crucial, since disease dynamics and public health policy vary by geography. Here, we introduce a new method for estimating levels and trends in local-level rates of disease-relevant contacts by combining information from national-level contact data and other locality-specific data sources. We illustrate our method by estimating novel state-level estimates of contact patterns in the US over the course of the COVID-19 pandemic. These estimates are an important resource for researchers and policymakers who wish to understand the state-level dynamics of the COVID-19 pandemic in the United

Framework Repository: https://doi.org/10.17605/OSF.IO/AECWN.

**Funding:** A Berkeley Population Center pilot grant (National Institute of Child Health and Human Development P2CHD073964) was awarded to ASM and DMF. DMF received additional funding from the Hellman Fellows Program. The funders had no role in study design, data collection and analysis, decision to publish, or preparation of the manuscript.

**Competing interests:** The authors have declared that no competing interests exist.

States. The new methods we introduce are applicable to any directly transmitted infectious disease and can be applied in many different settings all over the world.

This is a *PLOS Computational Biology* Methods paper.

## Introduction

As of April 2022, the COVID-19 pandemic is estimated to have infected over 505 million people and caused at least 6.2 million deaths around the world [1]. Like many other directly transmitted diseases—including influenza, pertussis, and measles—the pathogen that causes COVID-19 is spread through respiratory droplets that can be transmitted during close, person-to-person interactions [2–6]. For these directly transmitted infectious diseases, reliable quantitative estimates of levels and patterns of interpersonal contact are central to measuring, modeling, and reducing transmission. Contact patterns serve as critical inputs to mathematical disease models, allowing for the direct incorporation of contact heterogeneity by age, geography, and time to more realistically simulate disease transmission dynamics [7, 8].

While national-level estimates of age-specific contact patterns are available in many countries [6, 9–13], local-level estimates are almost never available. Yet these local-level contact patterns are crucial, since disease dynamics and public health policy vary by geography within a country. This need for timely local-level estimates of contact patterns is especially apparent during the COVID-19 pandemic. In the US, for example, state-to-state differences in physical distancing policies, epidemic waves, seasonality, culture, and other factors have led to many heterogeneous COVID-19 outbreaks, rather than one coherent national outbreak. To accurately model these local COVID-19 outbreaks, each state would ideally collect its own periodic contact surveys to quantify changing local patterns of interpersonal interaction over the course of the pandemic. However, repeatedly collecting contact data for each state would require a very large sample, making it prohibitively expensive.

To overcome this challenge, we introduce a novel approach to estimating subnational interpersonal contact patterns by combining information from national-level contact data and other locality-specific data sources—such as census data—using multilevel regression with poststratification (MRP) [14, 15]. Conceptually, our approach proceeds in two steps: first, we fit Bayesian hierarchical models to age-specific contact patterns observed in the national-level contact survey data. The purpose of these models is to predict age-specific contact rates based on geographic and demographic covariates. Second, we poststratify our model-based estimates by aggregating them up to state-level age-stratified contact rates using weights that represent the population composition of each state. Age-stratified contact rates are key because of age-related variation in transmission, mortality, and immunity [16, 17].

We apply our method to estimate daily age-stratified contact matrices for all 50 US states and Washington DC between April 2020 and May 2021. All estimates are publicly available at https://osf.io/aecwn/?view_only=0de9a7de36b241d1a5f3f7174629e368. Our results reveal substantial state-level heterogeneity in levels and trends of contacts across the US over the course of the COVID-19 pandemic, with overall contact intensity being highest for the Southern and Rocky Mountain states and lowest for states on the West Coast and in New England. Contact patterns generally became increasingly assortative with respect to age over our observation window, with significant state-level differences in these trends. Taken together, our results demonstrate that national-level contact patterns are not a suitable replacement for state-level contact patterns. Our method can be applied in many other settings where national-level

contact data is available, but local-level estimates are needed. The resulting age-stratified contact matrices are relevant for a range of infectious diseases, such as COVID-19, influenza, pertussis, and measles.

## Data and methods

### Data

We use data from Waves 1–6 of the Berkeley Interpersonal Contact Study (BICS), a nationally-representative study of interpersonal interaction in the US [11]. Respondents were asked to report on the number of household and non-household persons they had *conversational contact* with the day before the survey. Conversational contact was defined as a two-way conversation with three or more words in the physical presence of the other person. This is the definition used by the POLYMOD project [9]. Respondents were asked to provide age and sex information on all household members and detailed information on up to three of their non-household contacts, including demographic profile, length of interaction, and where the conversation took place. These detailed contacts were up-weighted to reflect the total number of contacts, for which detailed information may or may not have been collected (see Section 1.1 in S1 Appendix for details).

Six waves of data were collected from April 2020 to May 2021. Waves were generally conducted over a few weeks. Table 1 shows the number of interviews conducted per month.

### New approach to estimating subnational contact patterns

Our paper introduces a new method for estimating levels and trends in state-level rates of disease-relevant contacts in the United States over the course of the COVID-19 pandemic by combining information from national-level contact data and other locality-specific data sources. The methods we introduce are applicable to any directly transmitted infectious disease—including, for example, influenza, pertussis, and measles—and can be applied in many different settings all over the world.

Researchers construct age-structured contact matrices to summarize patterns of interpersonal contact in a population. These age-structured contact matrices describe average rates of contact between age groups and serve as a key input to most mathematical models of directly transmitted infectious diseases. Formally, we assume the population has been divided into $A$ discrete age groups. Let $c_{ij}$ be the average number of contacts that a person in age group $i$ has with people in age group $j$ over a given time period. Then the contact matrix $C = (c_{ij})$ is the $A \times A$ matrix that is composed of the expected number of contacts between each pair of age groups.

A growing body of methodological work is concerned with how best to estimate the entries $c_{ij}$ of the contact matrix $C$ [6, 9, 10, 12, 13, 18, 19]. A conventional strategy, called indirect

**Table 1. Number of BICS interviews administered monthly.**

| Wave | Month | Year | Interviews |
|---|---|---|---|
| 1 | April | 2020 | 856 |
| 1 | May | 2020 | 525 |
| 2 | June | 2020 | 1113 |
| 3 | September | 2020 | 1486 |
| 4 | December | 2020 | 1094 |
| 5 | February | 2021 | 1304 |
| 6 | May | 2021 | 2412 |

estimation, is based on fitting models that back-estimate the entries in the contact matrix $C$ from epidemiological data, such as case-counts or estimates of seroprevalence. Indirect estimation is appealing because these methods do not require special data collection. However, this convenience comes at a cost—indirect estimation requires making several assumptions that are hard to validate or check. More recently, attention has turned to direct estimation methods which involve collecting data to estimate each entry $c_{ij}$ of the contact matrix. Several different types of data can be collected to achieve this goal: previous studies have explored estimating $c_{ij}$ from time-use data [10, 18], wearable sensors [20, 21], and surveys, including diary studies [9, 11, 22].

This study focuses on using surveys that ask respondents to report about their face-to-face interactions with other people [6]. This direct estimation approach is appealing because it is extremely flexible. Surveys can be used by researchers to: (1) make inferences about contact patterns in a population that is not constrained to a small geographic area, (2) gather information about the interactions most relevant to the spread of directly transmitted pathogens, and (3) collect important information beyond the contacts themselves, including contextual information about interpersonal interactions (e.g., their location and duration) and about the people who interacted (e.g., their age, sex, occupation, and relationship to one another). This contextual information can be very useful when developing policies to limit contacts in the hopes of slowing the spread of a directly transmitted infectious disease.

While surveys or diary studies have now been used to produce national-level estimates of age-specific contact patterns in many countries around the world [6, 9–11, 23–25], the majority of countries still lack nationally representative contact surveys. One stream of research has developed an approach for extrapolating synthetic national-level contact matrices in countries where no contact data are available at all [12, 13]. Principled national-level extrapolations of contact patterns are vastly preferable to no information at all. These principled extrapolation methods can be seen as a complement to the methods we introduce here, which can instead be used when national-level contact data are available but researchers wish to make fine-grained, local-level inferences. Our approach is thus more similar to interpolation, as opposed to extrapolation.

In order to use nationally representative BICS data to estimate contact matrices at the state-level [26–31], we extend a promising approach called multilevel regression with poststratification (MRP). First developed by Gelman and Little (1997), MRP has recently been extended to estimate subnational public opinion [14, 15, 32, 33], forecast elections [34], and estimate small-area population health outcomes [35]. This study is the first application of MRP to the estimation of social contact patterns.

MRP consists of two main steps. In the first step (multilevel regression), we fit Bayesian hierarchical models to estimate contact patterns for all poststratification "cells"—mutually exclusive groups created by cross-classifying key covariates. For example, one cell used in our study is Non-Hispanic Black women aged 25–35 living in New York in a household of size 4. In the second step (poststratification), we aggregate the cell-level estimates up to the state-level by weighting each cell by its relative proportion in the state. For our study, we add a third step, which takes state-level estimates and aggregates them up to contact matrices.

MRP has traditionally been used to study static outcomes. However, we needed to estimate contact matrices for each US state over many time periods. One approach is to use a separate MRP model for each time period (e.g., year) and combine these separate MRP models into a single time-series model. However, combining separate MRP models doesn't make full use of all available information and is less flexible for incorporating time trends into a model. Therefore, we fit a single "dynamic MRP" model explicitly incorporating a time trend [15]. This model allows us to produce estimates for all 51 states and 411 different time periods (days). Dynamic MRP allows us to take advantage of all data available when making predictions.

To summarize, we follow three steps to estimate state-level contact matrices from nationally-representative BICS data: (1) we fit Bayesian hierarchical models to the national-level survey data; (2) we poststratify model estimates; (3) we calculate state-level contact matrices from the poststratified estimates. We now describe each of these steps in turn.

## Models

Our hierarchical models overcome the instability of estimates from very finely partitioned cells by estimating cell-level averages from a hierarchical model that allows for "random" or "modeled" effects for some predictors. Random effects allow for partial pooling towards the group mean, with greater pooling for less-populated cells [27]. When little to no data are available for a cell, estimates are based on data from similar cells; when lots of data are available for a cell, estimates are closer to the original survey responses.

There are fundamental differences between household contacts and non-household contacts; for example, we would expect non-household contacts to change substantially over the course of the pandemic while household contacts remain relatively stable. Therefore, we develop two flexible hierarchical models for the contact data in the BICS survey: one for non-household contacts and one for household contacts. We estimate the total number of contacts by summing the predicted number of household and non-household contacts.

For the non-household contacts models, we include a national-level cubic time trend and state-level hierarchical cubic time trend, which allows for state-level deviations from the national time trend. We chose our final model by fitting a series of models with different specifications and performing model selection using Pareto-smoothed importance sampling leave-one-out cross validation (PSIS LOO-CV) to select the model with the best performance [36]. For further details, see Section 2.3 in S1 Appendix.

We incorporate Google Community Mobility reports on visits to seven different place categories—grocery and pharmacy, parks, transit stations, retail and recreation, residential, and workplaces [37, 38]—into our model. We follow [39] by aggregating these seven signals into one predictor, the three-week rolling average of the first principal component. We find this mobility predictor only marginally increases our model's performance (see Table A in S1 Appendix). However, we include the mobility predictor in our model to highlight the model's ability to incorporate state-level contextual predictors. Highly comparable estimates from a model without the mobility predictor are presented in Section 3.4 in S1 Appendix, demonstrating this method can be used without mobility data.

For our model, we let $y_{ic}$ be the number of contacts person $i$ has with people in age category $c$ for $1 \leq c \leq C$. Then $\mathbf{y}_i = (y_{i1}, \ldots, y_{iC})^T$ is a vector with the number of contacts person $i$ has with people in each age group. We fit separate models to predict the total number of contacts (calculated by up-weighting detailed contacts) in each of the 7 defined age categories: 0–17, 18–24, 25–34, 35–44, 45–54, 55–64, and 65+. We assume each $y_{i,c}$ follows a negative binomial distribution,

$$y_{i,c} \sim \mathcal{NB}(\mu_{i,c}, \phi), \tag{1}$$

where $\mu_{ic}$ is the expected number of contacts. We model the log of the expected number of contacts using

$$\log(\mu_{i,c})_{non-household} = \alpha_{0,c} + a_{j[i],c}^{\text{gender}} + a_{j[i],c}^{\text{age category}} + a_{j[i],c}^{\text{household size}} + a_{j[i],c}^{\text{race}} + a_{j[i],c}^{\text{mobility}}$$

$$+ a_{j[i],c}^{\text{gender:agecat}} + a_{j[i],c}^{(\text{day}+\text{day}^2+\text{day}^3)} + \tag{2}$$

$$+ b_{j[i],c}^{(1+\text{day}+\text{day}^2+\text{day}^3)|\text{state}}$$

and

$$\log(\mu_{i,c})_{household} = \alpha_{0,c} + a_{j[i],c}^{\text{gender}} + a_{j[i],c}^{\text{age category}} + a_{j[i],c}^{\text{household size}} + a_{j[i],c}^{\text{race}} + a_{j[i],c}^{\text{gender:agecat}}$$
$$+ b_{j[i],c}^{1|\text{state}},$$

(3)

where $\alpha_{0,c}$ is the fixed intercept term and the terms $a_{j[i],c}^{\text{var}}$ denote the population-level effects (fixed effects) associated with each categorical variable. We use the subscript $j[i]$ to denote the group membership for the $i$th respondent. For example, the $a_{j[i],c}^{\text{race}}$ variable can take values $\{a_{Black,c}^{\text{race}}, a_{Hispanic,c}^{\text{race}}, a_{White,c}^{\text{race}}, a_{OtherRace,c}^{\text{race}}\}$ depending on the racial self-identification of respondent $i$. The terms $b_{j[i],c}^{\text{var}|\text{state}}$ denote the group effects (random effects or hierarchically modeled effects).

We include group effects because they allow the model to take advantage of partial pooling: when many observations are available, the estimates are closer to the original survey responses, and when few observations are available for a group, the estimates are pulled away from survey responses towards the overall mean. By assuming that the group effects for a given variable (e.g. the $b_{j[i],c}^{1|\text{state}}$ for the 51 states) are exchangeable, we can treat the variance $\sigma^{2,\text{state}}$ as a free parameter that will affect the degree of partial pooling. Thus, the amount of partial pooling—how much cell estimates are pulled towards the overall mean—is estimated from the data. In the household contact models, we model the state-specific intercepts as $b_{j[i],c}^{1|\text{state}} \sim N(0, \sigma^{2,\text{state}})$. We use a weakly informative hyper-prior for the variance parameters to avoid boundary estimates (i.e., to avoid a group-level variance estimate of 0 where such an estimate is a priori implausible [40, 41]). Specifically, we assume a half-t-distribution (truncated from below at 0) with 3 degrees of freedom for the prior of the standard deviation, $\sigma^{\text{state}}$: $\sigma^{\text{state}} \sim$ half-$t_3(0, 2.5)$.

In the non-household contact models, we have state-specific cubic time trends, which require estimating a vector of four parameters per state. To benefit from partial pooling, we model these four state-specific parameters hierarchically via

$$b_{j[i],c}^{(1+\text{day}+\text{day}^2+\text{day}^3)|\text{state}} \sim N(0, V),$$

(4)

where the variance-covariance $V$ is

$$V = \text{diag}(\sigma)\Sigma\text{diag}(\sigma)$$
$$\Sigma \sim \text{LKJ}(1)$$
$$\boldsymbol{\sigma} \sim \text{half-}t_3(0, 2.5).$$

(5)

Here, we employ the strategy of Bürkner (2017), which separately models the marginal variances $\boldsymbol{\sigma}$ (a vector with four entries) and the correlation structure between the four parameters. The LKJ(1) prior implies a uniform prior distribution over correlation matrices. Similar to the household contacts model, we adopt a weakly informative half-t-distribution hyperprior for the variances. Specifically, we use a half-t-distribution with 3 degrees of freedom (truncated from below at 0) for the standard deviation.

Finally, we specify relatively uninformative priors for some of the model parameters:

$$\phi \sim \text{Gamma}(\alpha = 0.01, \beta = 0.01)$$
$$\alpha_{0,c} \sim t_3(-2.3, 2.5).$$

(6)

All other parameters have a flat prior.

## Poststratification

For our post-stratification step, we aggregate the cell-level estimates up to the state-level by weighting each cell by its relative share of the state's population. We define the target population using the Public Use Micro Data Samples from the American Community Survey 2014–2018 5-year sample [42]. We split the target population into 14,688 cells based on 5 categorical variables: gender (2 categories), state (51 categories), age group (6 categories), household size (6 categories), and Race / Hispanic origin (4 categories). Note that, under traditional poststratification, our survey would not be large enough to produce estimates for these 14,688 cells: we would have little or no data for most cells, and the resulting estimates would be extremely noisy. However, this number of cells is not a problem for MRP, which borrows information from similar cells to smooth out estimates for otherwise noisy cells.

Fig 1 illustrates our method by comparing our MRP-adjusted estimates for April 2020 with the original survey responses for two states: California, where many interviews were conducted

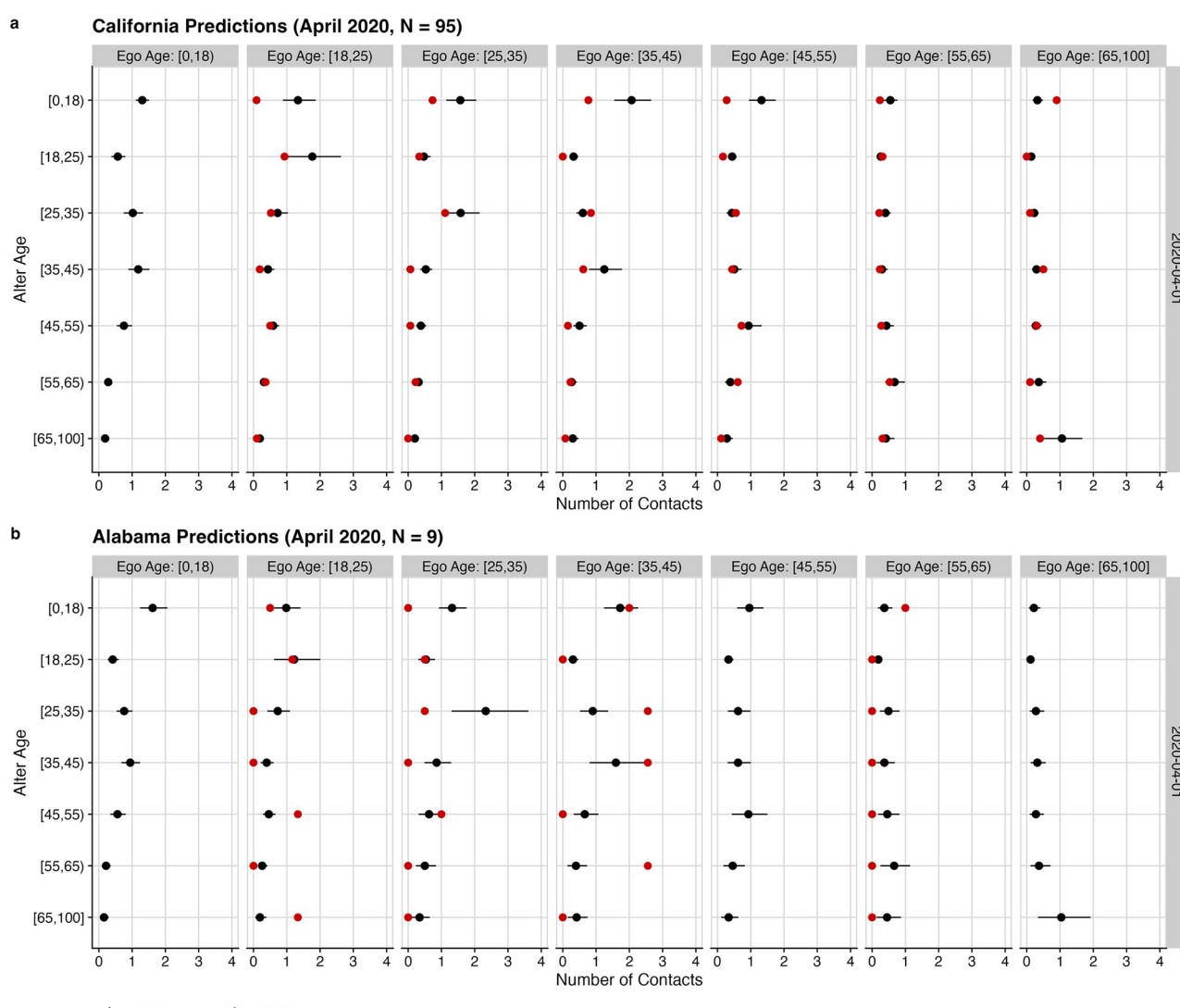

**Fig 1. Comparison of MRP-adjusted estimates (black) with unweighted BICS estimates for Alabama (panel A) and California (panel B) from April of 2020.** MRP estimates are presented with 95% credible intervals. Where more data are available, predictions more closely track original survey estimates.

(N = 95), and Alabama, where fewer interviews were conducted (N = 9). While the MRP estimates align closely with the original survey responses for California, they differ significantly for Alabama. This reflects the partial pooling of the hierarchical model: since Alabama has little data available, MRP estimates use information from similar cells to smooth out the noisy raw estimates produced from just 9 interviews. This partial pooling is particularly relevant for months where we collected no data. For example, no interviews were conducted in Alabama in November, but we are still able to use the dynamic MRP model to make predictions for November.

Next, we used the models to predict average age-specific contacts in each cell, and then poststratified the model predictions using the American Community Survey 2014–2018 5-year sample. The resulting estimates of state-level contact patterns for April 2020 to May 2021 were adjusted for reciprocity and arranged into contact matrices. As an illustration, Fig 2 shows the estimated contact matrices for South Dakota and Washington in July 2020. Section 3.6 in S1 Appendix shows the estimated contact matrices for all states and all months from April 2020 to May 2021.

For each estimated contact matrix $\widehat{C}$, we define the relative contact intensity index to be equal to $\frac{\rho(\widehat{C})}{\rho(C^{\text{UK}})}$, where $\rho(\widehat{C})$ is the dominant eigenvalue of the estimated contact matrix and $\rho(C^{\text{UK}})$ is the dominant eigenvalue of the contact matrix for the UK POLYMOD study. We follow other studies in using contact patterns from UK participants in the POLYMOD study as a pre-pandemic baseline reference [9, 11]. Assuming no age-specific differences in susceptibility or infectiousness, the contact intensity index is directly proportional to the ratio of the $R_0$ values implied by the two contact matrices [43]. Additionally, we calculate each age group's relative contribution to overall transmission. This is proportional to the leading left eigenvector of

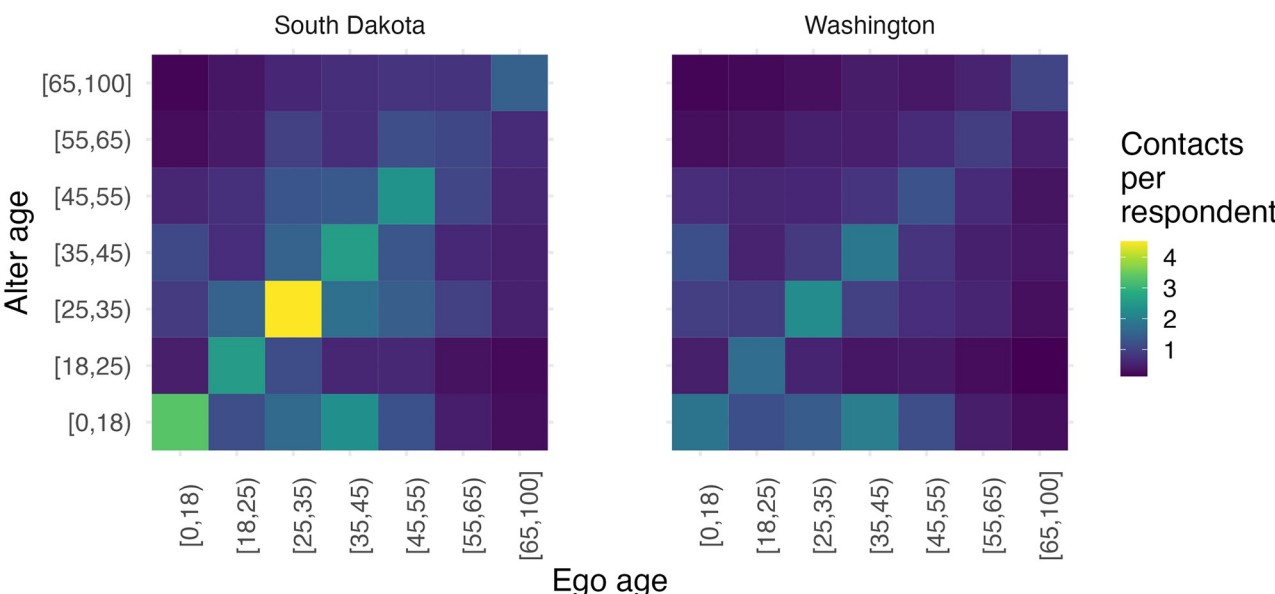

**Fig 2. MRP estimates of state-level contact matrices for Washington and South Dakota in July, 2020.** Each entry *i, j* in the contact matrix represents the average number of contacts that a person (ego) in age group i has with other people (alter) in age group j in the past 24 hours. Each cell is shaded according to the estimated mean number of total (household and non-household) contacts. The contact matrices have been adjusted for reciprocity; however, this is not necessarily reflected visually in the contact matrices due to the differential size of the population subgroups.

the contact matrix, assuming no age-specific variation in transmissibility or susceptibility [44]. The leading left eigenvector is defined as a row vector $\mathbf{v}_L$ such that $\mathbf{v}_L \widehat{C} = \rho(\widehat{C})\mathbf{v}_L$, where $\widehat{C}$ is the estimated contact matrix and $\rho(\widehat{C})$ is the leading eigenvalue.

Finally, to assess the degree of assortative mixing with respect to age (i.e., the tendency for people to have higher contact with others similar to them in age), we estimate the q-index. The q-index gives a measure of departure from perfectly proportionate mixing: a value of 0 indicates perfectly proportionate mixing and a value of 1 indicates a full assortativity. Specifically, the q-index is defined as:

$$\text{q-index} = \frac{|\widehat{\lambda_2}|}{|\widehat{\lambda_1}|} \tag{7}$$

where $\widehat{\lambda_1}$ is the largest eigenvalue and $\widehat{\lambda_2}$ is the second largest eigenvalue.

We use our model to estimate daily age-stratified contact matrices for all 50 US states and for Washington DC. The model combines information from (1) the Berkeley Interpersonal Contact Study (BICS) waves 1—6 (N = 8,790), which collected responses at the US national level from April 2020 to May 2021; and (2) locality-specific data sources, including the American Community Survey and digital estimates of population mobility from Google ("Methods"). Our model produces estimated age-stratified contact rates for each state and each day, but we focus here on monthly average estimates for each state in order to investigate changes over moderate time scales and to smooth predicted values, as recommended in the small-area estimation literature [45, 46].

Fig 2 compares predicted contact matrices for Washington and South Dakota—two states chosen to illustrate an example of a state with low and high contact, respectively—over all days in July 2020. Section 3.6 in S1 Appendix shows additional estimated contact matrices for all states and all months from April 2020 to May 2021. Two substantive findings emerge from Fig 2 and Section 3.6 in S1 Appendix. First, the levels of interpersonal contact differ significantly across states. For example, in July 2020, members of the [18–25) age group in South Dakota had on average 11.6 total contacts in the past 24 hours, while members of the [18–25) age group in Washington had only 6.4 total contacts in the past 24 hours. Second, estimated state-level contact patterns have high levels of assortative mixing by age for all states and across the course of the pandemic: at each month, the leading diagonal stripes visible in both state's estimated contact matrix in Fig 2 show highest relative rates of contact between members of the same age group at all or nearly all ages. Members of the [0–18) age group are also estimated to have high average daily contacts with members of age groups [25–35) and [35–45) across all states and time periods, suggesting the persistent influence of generational and household structure on interpersonal interaction. These findings are consistent with previous national-level contact studies conducted in countries outside the US both before and during the COVID-19 pandemic [9, 47].

To compare the intensity of interpersonal contact across states and time, we define a quantity called the *relative contact intensity index*. The relative contact intensity index summarizes how high or low age-structured contact rates are in a specific state at a specific time, relative to a baseline. A relative contact intensity value of 1 means that contact intensity is unchanged since baseline; a value less than 1 means that contact intensity has reduced since baseline, and a value greater than 1 means that contact intensity has increased since baseline (for full mathematical details, see "Methods").

Fig 3 shows how the relative contact intensity index for each state changes from March of 2020 to May of 2021, using UK participants in the POLYMOD study as a pre-pandemic

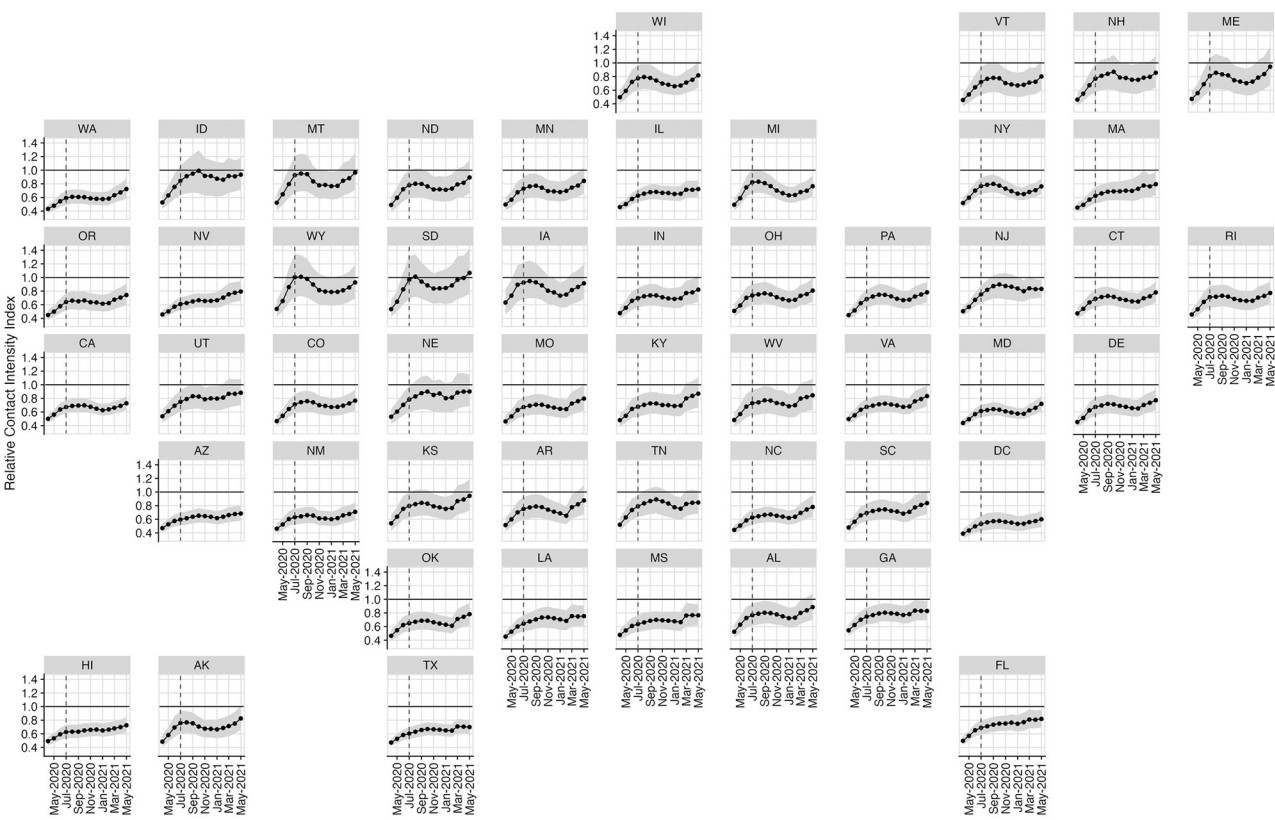

**Fig 3. Monthly state-specific estimates of relative contact intensity index over time.** Horizontal black line shows pre-pandemic baseline contact intensity. Vertical dashed line denotes July 2020, the month we present contact matrices for in Fig 2.

baseline reference [9]. Consistent with the cross-state variation Fig 2 revealed for July 2020, Fig 3 shows significant heterogeneity in total levels of contact across states at all time periods. This heterogeneity suggests that national-level contact rates would be a misleading proxy for the local level: in July 2020, national contact rates would over-estimate contact intensity for Washington by 18%, while under-estimating contact intensity in South Dakota by 34%. All states experienced an upward trend in contacts over time. Relative contact intensity is lowest in March 2020 and increases until August 2020. Beginning in September 2020, some states experience a substantial decline in contact intensity (Iowa, Kansas, and New York), while other states (Florida, Hawaii, and Massachusetts) experienced a modest increase. In January 2021, contact intensity began increasing in almost every state until the end of our observation window in May 2021. The variation in contact intensity was lowest in April 2020, and peaked in August 2020 (see Fig D in S1 Appendix for full analysis).

The relative contact intensity index is a useful measure of the overall intensity of interpersonal contact. But, by summarizing the level of contact across all age groups with a single quantity, the relative contact intensity index can mask important age-specific variation over time. For example, because many diseases such as COVID-19 and influenza have strong age-specific patterns of mortality [16], age-specific contact rates are key to understanding underlying transmission dynamics. To demonstrate the implications of age-specific heterogeneity in contact rates, we create an index capturing each age group's relative contribution to overall transmission, assuming no age-specific differences in susceptibility or infectiousness [48]. Fig 4 shows the results for each age group in three large US states with different population and

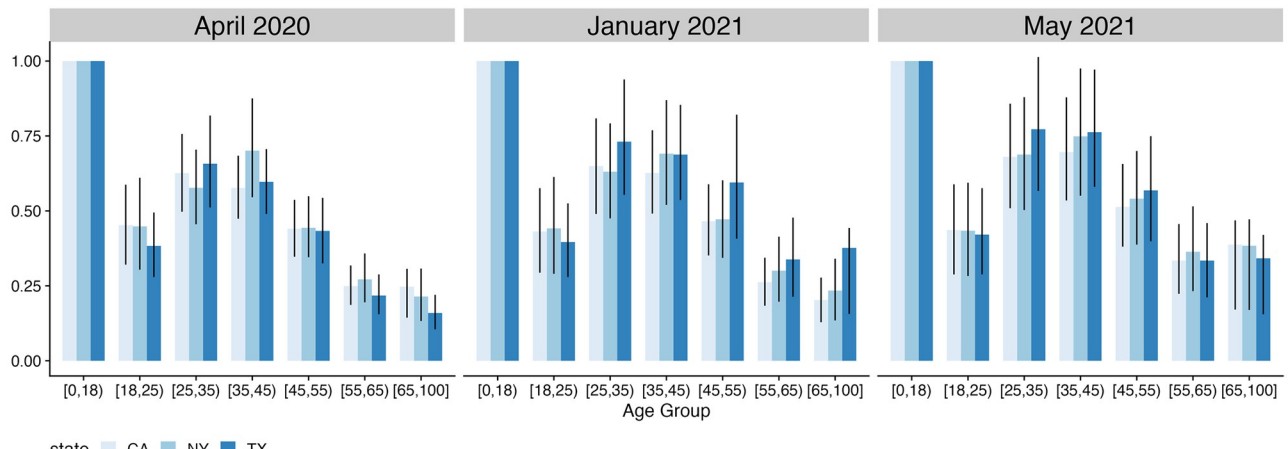

**Fig 4. Relative contribution of each age-group to overall transmission in California, New York, and Texas assuming no age-specific differences in transmissibility or susceptibility.** We pick the youngest age group [0–18] as a reference group which all other estimates are rescaled to. Error bars represent 80% credible intervals.

policy dynamics: California, New York, and Texas. For interpretability, we normalize each value by dividing by the value for the [0–18) age group within each state and month. The contribution to overall transmission is highest for the [0–18) age group and lowest for the [55–65) and [65–100) age groups. These differences vary across states. For example, in January 2021, the relative contribution for the [65–100] age group was 87% larger for Texas than for California.

Finally, to understand how assortativity with respect to age changed over time, we estimate monthly, state-level estimates of the q-index. The q-index takes on values between 0 (proportionate mixing) and 1 (fully assortative mixing) [49]; see "Methods" for full mathematical definition. Fig 5 shows contacts generally became more age-assortative over time, consistent with previous findings in Wuhan, China [49]. However, trends vary across states: New Jersey's age assortativity peaks in January 2021, with a gradual decline until May of 2021. In California, age assortativity remains low until January 2021, when there is a sharp uptick.

Previous work has found that non-pharmaceutical interventions (NPIs), such as state-level social distancing regulations and their enforcement, are associated lower aggregate-level contact intensity [25, 50–53]. As a robustness check, we refit our models adding the Oxford Stringency Score as a covariate [54]. The Oxford Stringency Score in an NPI that measures the stringency of "lock-down" policies that primarily restrict people's social behavior. We find no significant improvement the predictive accuracy of the model (see Table B in S1 Appendix). Incorporating data on NPIs in other settings may be very useful in predicting contact rates, and we emphasize that our modeling framework makes this possible.

## Discussion

In this paper, we introduce a dynamic MRP model for estimating state-level contact patterns from national-level survey data. We apply this model to national-level contact survey data collected from the Berkeley Interpersonal Contact study to produce age-stratified contact matrices for all 50 US states and Washington DC. This method for estimating subnational contact patterns can be easily adopted to other settings in countries that have recently collected national-level contact data and have available poststratification targets from a census or similar data source.

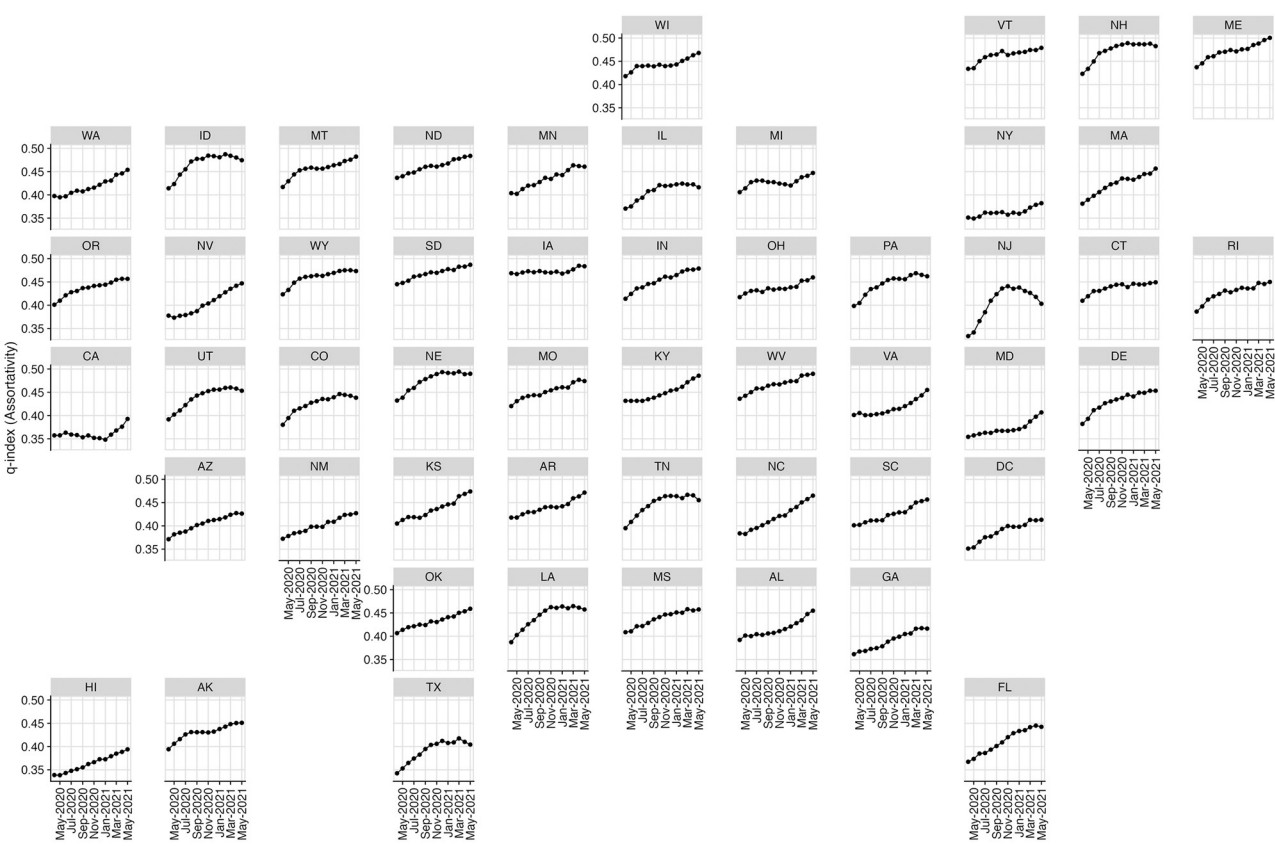

**Fig 5. Change in age assortativity, as measured by the q-index, over the course of the pandemic.** A higher q-index value corresponds to higher age assortativity.

Our results reveal several key substantive insights into social contact patterns in the US during the COVID-19 pandemic. First, there is significant heterogeneity across states in overall levels of contact intensity. Second, while trends varied by state, contact intensity changed dramatically over time, with a low in April 2020, a spike in the summer of 2020 followed by a decline until January of 2021, and a gradual increase between February 2021 and May 2021. Finally, we observe a general increase in age-related contact assortativity over time, again with distinct state-level patterns of change. In sum, our findings demonstrate the crucial importance of state-level contact patterns; we find that during the COVID-19 pandemic in the US, national-level contact patterns mask important state-level variation and are not a suitable proxy for state-level estimates.

We have several methodological recommendations for other studies that may wish to produce subnational contact rate estimates from national contact data. First, we recommend researchers collect data on both household and non-household contacts and model them separately, as predictors for household and non-household contacts may differ in important ways. Second, if the contact data was collected over time, a time trend should be included in the model [15]. The cubic time-trend specified in our model may not generalize to other situations. Third, we recommend researchers fit several models with different time trends and use principled model selection techniques to pick the final model. Finally, we echo past advice in the small-area literature: model at the finest level possible and aggregate to the desired geography and time scale [45, 46].

Despite the promise of this approach, our study has important limitations and there are several directions for future work. We employ an ad-hoc approach to enforcing reciprocity in aggregate contacts between age groups (see "Methods"). Future work could explicitly incorporate these reciprocity constraints into the modeling approach. This would allow for a fully Bayesian strategy with credible intervals that may better capture the full uncertainty inherent in our model. The model could be adapted to more directly capture the age structure of reported contacts. This will involve more explicitly modeling the relationship between reported contacts in different age groups (as opposed to fitting parallel negative binomial models, as we did here). Because we have no survey respondents under the age of 18, we use the principle of reciprocity to impute the average daily number of contacts for this age group [11]; future surveys could try to interview children under 18 or obtain proxy reports from children's parents. The reported contacts by age could be smoothed within the modeling framework, thereby reducing large jumps between neighboring cells in the total number of contacts. The BICS survey did not directly interview respondents under age 18; thus, we follow previous studies in inferring the within-group contact from all (excluding school) contacts from the UK POLYMOD (see Section 1.2 in S1 Appendix for details). As some schools in the US were open during this period, our estimate for the relative contribution the [0–18) group makes to overall transmission may be an underestimate for some states and time periods. Additionally, we assumed that all household members were contacts, but future research could relax this assumption and model trends in contact among household members over time. Perhaps most importantly, these results could be validated against other estimates from a high-quality state-level contact study, which ideally captures several states during several different time periods. These are important components of future work, which will advance our ability to produce accurate and timely small-area estimates of contact patterns from national-level contact data.

## Supporting information

**S1 Appendix. File containing additional detail about the data, model, and supplemental tables and figures.** Fig A. The $R_0$ value implied by contact patterns in each state. Fig B. State-specific MRP estimate of total contact (combined household and non-household contacts). Fig C. Illustration of partial pooling. Fig D. Monthly standard deviation of contact intensity across states. Fig E. Model input and output for state of California. Fig F. State-level change in relative contact intensity index over time from a model without mobility data predictors. Fig G—Fig T. Estimated State-level Contact Matrices by month. Table A. Model selection metrics. Table B. Model performance with and without the Oxford Stringency Score. (PDF)

## Acknowledgments

For helpful feedback on earlier versions of the manuscript, the authors would like to thank Sophia Rabe-Hesketh, Monica Alexander, Carl Schmertmann, and members of the Population Association of America 2021 session "Bayesian Demography."

## Author Contributions

**Conceptualization:** Dennis M. Feehan.

**Data curation:** Ayesha S. Mahmud, Dennis M. Feehan.

**Formal analysis:** Casey F. Breen.

**Funding acquisition:** Ayesha S. Mahmud, Dennis M. Feehan.

**Investigation:** Casey F. Breen.

**Methodology:** Casey F. Breen.

**Project administration:** Dennis M. Feehan.

**Software:** Casey F. Breen.

**Supervision:** Ayesha S. Mahmud, Dennis M. Feehan.

**Visualization:** Casey F. Breen.

**Writing – original draft:** Casey F. Breen.

**Writing – review & editing:** Casey F. Breen, Ayesha S. Mahmud, Dennis M. Feehan.

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
