## [Decision Letter · Decision Letter 0]

17 Jun 2022

Dear Mr. Breen,

Thank you very much for submitting your manuscript "Novel Estimates Reveal Subnational Heterogeneities in Disease-Relevant Contact Patterns in the United States" for consideration at PLOS Computational Biology.

As with all papers reviewed by the journal, your manuscript was reviewed by members of the editorial board and by several independent reviewers. In light of the reviews (below this email), we would like to invite the resubmission of a significantly-revised version that takes into account the reviewers' comments.

All three reviewers found that this submission represents a substantial, interesting, and valuable contribution. They also raised a number of constructive points that could strengthen the paper, and, were those to be addressed, I would be excited to consider a revised submission. One common thread raised by R1 and R2 is validation. Here, the reviewers make substantially different suggestions; I suggest that the authors consider whether the constructive goals of both reviewers might be reasonably achieved even if only one approach to validation is followed.

We cannot make any decision about publication until we have seen the revised manuscript and your response to the reviewers' comments. Your revised manuscript is also likely to be sent to reviewers for further evaluation.

Sincerely,

Daniel B Larremore, Ph.D.

Associate Editor

PLOS Computational Biology

Virginia Pitzer

Deputy Editor-in-Chief

PLOS Computational Biology

All three reviewers found that this submission represents a substantial, interesting, and valuable contribution. They also raised a number of constructive points that could strengthen the paper, and, were those to be addressed, I would be excited to consider a revised submission. One common thread raised by R1 and R2 is validation. Here, the reviewers make substantially different suggestions; I suggest that the authors consider whether the constructive goals of both reviewers might be reasonably achieved even if only one approach to validation is followed.

Reviewer's Responses to Questions

**Comments to the Authors:**

Reviewer #1: The manuscript “Novel Estimates Reveal Subnational Heterogeneities in Disease-Relevant Contact Patterns in the United States” presents a novel methodology to estimate age-specific contact patterns at the subnational level. The method is based on multilevel regression with post stratification and can flexibly account for cell-level estimates with a very low number of counts. I think that this work has a good potential and addresses an important open question, that is the one to estimate sub-national social contact data starting from national surveys.

However, I think that there are several methodological issues that need to be addressed more carefully.

The approach of the authors uses a flexible level of pooling, with estimates that are closer to the survey response when many observations are available and estimates that are pulled away from survey response towards the mean when less data is available. This is indeed a very interesting and promising approach but I think that the authors are neglecting to consider the effect of Non Pharmaceutical Interventions (NPIs) on social contacts. Several studies [Gimma et al. 2020,Coletti et al. 2020, Liu et al. 2021] have shown that social distancing directions can reduce the number of social contacts substantially. It would seem that the hierarchical model is pooling data not taking the effect of NPIs into account as these are not included in the hierarchical model. Note that the inclusion of Google mobility is not sufficient to account for this, as Google mobility is not able to capture the full extent of contact reduction [Tomori et al. 2021]. This is even more important when looking at the contacts for the younger age class (0-18): When schools are closed, pooling information from situations in which schools are open (or vice-versa) would most likely produce wrong estimates. There are several possibilities to include the NPI In the model, for example including the Oxford stringency index as a possible covariate of the model (Oxford 2020), to avoid pooling together information that is not homogeneous with respect to NPIs.

Also, the authors mention that they develop a dynamical model that pulls information from different time points. Although this has the advantage of making use of the full information it is not clear to me whether there is any preference among pulling “over time” or pulling “over demographic strata”. To be more clear, to estimate data for women aged 25-35 living in New York in April, does the model prefer information from different time points (e.g.women aged 25-35 living in New York in July) or from different locations (women aged 25-35 living in New Jersey in April)? If this is selected from the model, it should be clarified and discussed also in relation to the previous point (different NPIs taking place over time in different locations).

Again related to the younger age class it seems to me that the authors are relying on several assumptions to estimate the contacts of children. I think that an interesting sensitivity analysis would be to exclude the 0-18 age class from the model, in order to estimate contact matrices for adults only and assess to what extent a “zero assumption analysis” differs from the main one.

The authors assume no age-specific differences in susceptibility and infectivity. This has been proven to be a wrong assumption by several works [Davies et al. 2021, Franco et al. 2021] and I think that a proper sensitivity analysis with respect to this assumption would be needed. Even better, the authors may drop this assumption and include one (or more, as sensitivity analysis) set of age-specific values for susceptibility and infectivity.

Finally, I think that some validation of the method would be useful. Although already used in other settings, MRP has never been applied to social contact data. A sensitivity analysis aimed at quantifying the impact of MRP for different levels of data sparsity would help the reader to appreciate the strength of the method. This could be done, for example, generating from the original dataset a subsample that for one given state and time point includes only (e.g.) 20%, 40%, 60% and 80% of the original data. Applying the MRP to these subsamples may help the reader to have an idea of the impact of pooling. Also, having that the pooled results for low percentage sub-sample (i.e. including just a fraction of the original data) are in line with the full data would provide a good sanity check of the method.

I also have some minor remarks, that can easily addressed:

Page 13, line 214: “the majority of countries lack nationally representative contact surveys”. Although this is true, a large number of new surveys have been developed in recent years in developed [Hoang et al.2019] as well as developing [Mousa et al. 2021]. Also data for covid 19 have been extensively collected [Liu et al. 2019]. The reader should be made aware of this large number of social contact studies available.

Page 14 Line 262: “while household contacts remain relatively stable.” In general, household contacts tend to go down when social distancing rules are relaxed. Could the authors take into account this trend as well?

Page 15 Line 284 “weighted”: do the authors here mean the weights of the post-stratification?

In conclusion, I think that the authors should address these issues, adapt the methodology accordingly, and re-submit their work.

Bibliography

Coletti et al:”CoMix: comparing mixing patterns in the Belgian population during and after lockdown ”. Scientific reports 2020.

Davies et al: “Age-dependent effects in the transmission and control of COVID-19 epidemics”. Nature Medicine 2020.

Franco et al:”Inferring age-specific differences in susceptibility to and infectiousness upon SARS-CoV-2 infection based on Belgian social contact data”. PLOS Computational Biology 2022.

Hoang et al:”A Systematic Review of Social Contact Surveys to Inform Transmission Models of Close-contact Infections”. Epidemiology 2019.

Gimma et al: “Changes in social contacts in England during the COVID-19 pandemic between March 2020 and March 2021 as measured by the CoMix survey: A repeated cross-sectional study”. PLOS Medicine 2022.

Hoang :

Liu et al: “Rapid Review of Social Contact Patterns During the COVID-19 Pandemic ”. Epidemiology 2021.

Mousa et al: “Social contact patterns and implications for infectious disease transmission – a systematic review and meta-analysis of contact surveys ”. ELife 2021.

Tomori et al: “Individual social contact data and population mobility data as early markers of SARS-CoV-2 transmission dynamics during the first wave in Germany-an analysis based on the COVIMOD study”. BMC Medicine 2021.

Reviewer #2: This article is both timely and informative, employing a sophisticated analysis that ties together a number of strands in MRP research in a way that is very useful for other researchers working on similar problems. I have only a few minor suggestions for improvement:

1. It would be useful to discuss ways in which the results might be validated. For example, could the household age-structure estimates be meaningfully compared with household age-structure estimates taken directly from the ACS? Even if such validation is not included in the article, it would be useful to discuss how this might be approached in future research.

2. I find Equation 9 in the supplemental materials (section 6.2) very confusing the way it is written and the way the variables are defined. I realized that this equation is being used in a lot of the contact matrix literature, but it is not clear to me why the i’s and j’s are used differently on the left-hand side (chatij is mean contacts of age group i with members of age group j) and the right hand side (where cji is used for this and where Nj is defined as the size of age-group i). Would it not be easier to interpret as:

chatij = 1/2Ni * (cijNi + cjiNj)

Where chatij and cij are both defined in terms of mean contacts for group i with group j, Ni is defined as the size of group i, etc.? There may be a good reason for writing it the you have it, but I would at least consider the advantages of writing it in this more intuitive way.

Reviewer #3: Thank you for the opportunity to review this manuscript.

The authors identify an important issue regarding the use of contact studies for the modelling of infectious diseases: there is often substantial geographical heterogeneity in contact behaviour which is not reflected in national-level contact surveys.

The authors present a technique (MRP) which has been successfully used in other fields and apply it to national contact studies to obtain sub-national contact estimates in the context of the COVID-19 pandemic in the USA.

The research is mostly sound and well-presented, but would benefit from a few additional improvements before it is suitable for publication.

Note that I am not an expert in hierarchical Bayesian modelling, so I cannot provide extensive comments on the use of this method here.

Comments:

- Abstract:

* "To overcome this challenge, we introduce a flexible model that can estimate age-specific contact patterns at the subnational level by combining national-level interpersonal contact data with other locality-specific data sources": if the word count for the abstract allows it, it would be good to refer to the methods you will be using here.

- Introduction:

* Line 16: I don’t think the heading ‘author summary’ belongs here.

* Line 19: “COVID-19 is spread through respiratory droplets that can be transmitted during close, person-to-person interactions”: Please add a reference which states that this is indeed the case for SARS-CoV-2, as all references here pre-date the pandemic.

- Results:

* Figure 1 (caption): Please define the abbreviation 'MRP' in the main text before using it.

* Line 75: Is there a specific reason Washington and South-Dakota were chosen to focus on here?

* Figure 2: Would it be possible to think of another way to present this which would make it easier to see the differences between states (although arranging the plots in the shape of the map of the US is a fun idea)?

*Line 119: Could you explain why you use relative R0 instead of calculating R0 outright - as this would be easier to interpret than using a relative R0 and then scaling it to the relative R0 of the youngest age group (which is a bit weird as well since this age group lacks sufficient data)?

* Figure 4: Same comment as Figure 2

* Line 135: “Understanding why we observe these differential state-level age assortativity trends is an important topic for future research.” : This statement may be better suited for the 'Discussion' section.

- Discussion:

* Line 166: “Finally, the reported contacts by age could be smoothed within the modeling framework.”: Could you explain further what is meant by this?

- Methods:

* I think it would be good to rewrite the Methods section for additional clarity (especially subsection 4.2): it would be better to be very clear about what is novel about what you are doing (i.e. the use of this technique in the context of contact matrices + the extensions you are adding to the technique), before going further into detail.

* Line 187: “this matrix”: what matrix?

* Line 282: 'C' is not defined (although it is easy to guess what it refers to).

* Line 345: Figure 1 does not show all states but only 2 of them.

- Supplemental Information:

* 'Guidance to Researchers': This section is not referred to in the text, and it is not very clear what the purpose of the section here is.

* Line 686: “We calculate R0 assuming a baseline R0 value of 2.54 for pre-pandemic contact rates.”: Could you add a reference to justify the use of this number?

**Have the authors made all data and (if applicable) computational code underlying the findings in their manuscript fully available?**

Reviewer #1: Yes

Reviewer #2: Yes

Reviewer #3: Yes

PLOS authors have the option to publish the peer review history of their article (what does this mean?). If published, this will include your full peer review and any attached files.

Reviewer #1: No

Reviewer #2: No

Reviewer #3: No
---

## [Decision Letter · Decision Letter 1]

27 Sep 2022

Dear Mr. Breen,

Thank you very much for submitting your manuscript "Novel Estimates Reveal Subnational Heterogeneities in Disease-Relevant Contact Patterns in the United States" for consideration at PLOS Computational Biology. As with all papers reviewed by the journal, your manuscript was reviewed by members of the editorial board and by several independent reviewers. The reviewers appreciated the attention to an important topic. Based on the reviews, we are likely to accept this manuscript for publication, providing that you modify the manuscript according to the review recommendations.

As you'll see below, there remain only some light suggestions to discuss the impact of NPIs in your work in relation to the broader literature, but no broader or methodological changes.  

Sincerely,

Daniel B Larremore, Ph.D.

Academic Editor

PLOS Computational Biology

Virginia Pitzer

Section Editor

PLOS Computational Biology

[LINK]

Reviewer's Responses to Questions

**Comments to the Authors:**

Reviewer #1: I thank the authors for having addressed my comments, especially the additional sensitivity analysis. However I think that the inclusion of the Non-Pharmaceutical-Intervention (NPI) in the model (included in the additional material and never mentioned in the text) should be discussed in the main text. A vast majority of works [Gimma et al. 2020,Coletti et al. 2020, Liu et al. 2021, Latsubaia et al. 2020, Tizzani et al. 2022] have shown a very strong effect of NPIs on the number of contacts. As such, the result of NPIs reducing the performance of the model is quite surprising and warrants further discussion in the main paper. In such a way, the reader is aware that these estimates of contact patterns are against the trend measured in these studies.

References:

Coletti et al: “CoMix: comparing mixing patterns in the Belgian population during and after lockdown ”.

Scientific reports 2020.

Gimma et al: “Changes in social contacts in England during the COVID-19 pandemic between March

2020 and March 2021 as measured by the CoMix survey: A repeated cross-sectional study”. PLOS

Medicine 2022.

Liu et al: “Rapid Review of Social Contact Patterns During the COVID-19 Pandemic ”.Epidemiology 2021.

Latsuzbaia A, Herold M, Bertemes JP, Mossong J (2020) Evolving social contact patterns during the COVID-19 crisis in Luxembourg. PLOS ONE 15(8): e0237128. https://doi.org/10.1371/journal.pone.0237128

Tizzani et al: Impact of tiered measures on social contact and mixing patterns in Italy during the second wave of COVID-19. https://doi.org/10.21203/rs.3.rs-1892693/v1

Reviewer #2: This new version addresses all of my suggestions. The internal validation check in S3.1 and the short discussion of validation on page 21 are both very helpful.

**Have the authors made all data and (if applicable) computational code underlying the findings in their manuscript fully available?**

Reviewer #1: Yes

Reviewer #2: Yes

PLOS authors have the option to publish the peer review history of their article (what does this mean?). If published, this will include your full peer review and any attached files.

Reviewer #1: No

Reviewer #2: No

Figure Files:

Data Requirements:

Reproducibility:

References:

---

## [Editor Report · Decision Letter 2]

16 Nov 2022

Dear Mr. Breen,

We are pleased to inform you that your manuscript 'Novel Estimates Reveal Subnational Heterogeneities in Disease-Relevant Contact Patterns in the United States' has been provisionally accepted for publication in PLOS Computational Biology.

Best regards,

Daniel B Larremore, Ph.D.

Academic Editor

PLOS Computational Biology

Virginia Pitzer

Section Editor

PLOS Computational Biology

---

## [Editor Report · Acceptance letter]

29 Nov 2022

PCOMPBIOL-D-22-00658R2 

Novel Estimates Reveal Subnational Heterogeneities in Disease-Relevant Contact Patterns in the United States

Dear Dr Breen,

I am pleased to inform you that your manuscript has been formally accepted for publication in PLOS Computational Biology. Your manuscript is now with our production department and you will be notified of the publication date in due course.

With kind regards,

Katalin Szabo
